# Interpretable agent communication from scratch (with a generic visual processor emerging on the side)

**Roberto Dessì**
Facebook AI Research
Universitat Pompeu Fabra
rdessi@fb.com

**Eugene Kharitonov**
Facebook AI Research
kharitonov@fb.com

**Marco Baroni**
Facebook AI Research
Universitat Pompeu Fabra
ICREA
mbaroni@fb.com

## Abstract

As deep networks begin to be deployed as autonomous agents, the issue of how they can communicate with each other becomes important. Here, we train two deep nets from scratch to perform large-scale referent identification through unsupervised emergent communication. We show that the partially interpretable emergent protocol allows the nets to successfully communicate even about object classes they did not see at training time. The visual representations induced as a by-product of our training regime, moreover, when re-used as generic visual features, show comparable quality to a recent self-supervised learning model. Our results provide concrete evidence of the viability of (interpretable) emergent deep net communication in a more realistic scenario than previously considered, as well as establishing an intriguing link between this field and self-supervised visual learning. [1]

## 1 Introduction

As deep networks become more effective at solving specialized tasks, there has been interest in letting them develop a language-like communication protocol so that they can flexibly interact to address joint tasks [1]. One line of work within this tradition has focused on what is arguably the most basic function of language, namely to point out, or *refer to*, objects through discrete symbols. Such ability would for example allow deep-net-controlled agents, such as self-driving cars, to inform each other about the presence and nature of potentially dangerous objects, besides being a basic requirement to support more advanced capabilities (e.g., denoting relations between objects).

While discreteness is not a necessary prerequisite for agent communication [2, 3], practical and ethical problems might arise if communication is incomprehensible to humans. A discrete code analogous to language is certainly easier to decode for us, helping us to understand the agents' decisions, and ultimately contributing to the larger goal of explainable AI [4].

In this paper, we study emergent discrete referential communication between two deep network agents that are trained from scratch on the task. We observe that the referential discrimination task played by the networks is closely related to pretext contrastive objectives used in self-supervised visual representation learning [5–8]. We exploit this insight to develop a robust end-to-end variant of a communication game. Our experiments confirm that, in our setup: i) the nets develop a set of discrete symbols allowing them to successfully discriminate objects in natural images, including novel ones that were not shown during training; ii) these symbols denote partially interpretable categories, so that their emergence can be seen as a first step towards fully unsupervised image annotation; iii) the visual representations induced as a by-product can be used as high-quality general-purpose features,

---

[1]Code: https://github.com/facebookresearch/EGG/tree/master/egg/zoo/emcom_as_ssl.

whose performance in various object classification tasks is not lagging much behind that of features induced by a popular self-supervised representation method specifically designed for this task.

## 2 Background

**Deep net emergent communication** There has recently been interest in letting deep nets communicate through learned protocols. This line of work has addressed various challenges, such as communication in a dynamic environment or how to interface the emergent protocol with natural language (see [1] for a survey). Probably the most widely studied aspect of emergent communication is the ability of deep net agents to use the protocol to refer to objects in their environment [e.g., 9–16].

The typical setup is that of a referential, or discriminative, communication game. In the simplest scenario, which we adopt here, an agent, the Sender, sees one input (the target) and it sends a discrete symbol to another agent, the Receiver, that sees an array of items including the target, and has to point to the latter for communication to be deemed successful. Importantly, task success is the only training objective; the communication protocol emerges purely as a by-product of game-playing, without any direct supervision on the symbol-transmission channel.

In one of the earliest papers in this line of research, Lazaridou et al. [9] used images from ImageNet [17] as input to the discrimination game; Havrylov and Titov [11] used MSCOCO [18]; and Evtimova et al. [12] used animal images from Flickr. While they relied on natural images, all these studies were limited to small sets of carefully selected object categories. Moreover, in all these works, the agents processed images with convolutional networks pretrained on supervised object recognition. While this sped up learning, it also meant that all the proposed systems *de facto* relied on the large amount of human annotated data used for object recognition training. Lazaridou et al. [13] and Choi et al. [14] dispensed with pre-trained CNNs, but they used synthetically generated geometric shapes as inputs.

Results on the interpretability of symbols in games with realistic inputs have generally been mixed. Indeed, Bouchacourt and Baroni [19] showed that, after training Lazaridou et al. [9]'s networks on real pictures, the networks could use the learned protocol to successfully communicate about blobs of Gaussian noise, suggesting that their code (also) denoted low-level image features, differently from the general semantic categories that words in human language refer to. In part for this reason, recent work tends to focus on controlled symbolic inputs, where it is easier to detect degenerate communication strategies [e.g., 10, 15, 16].

**Learning discrete latent representations with variational autoencoders** Like emergent communication, variational autoencoders and related techniques have been used to induce discrete variables without direct supervision [20–22]. There are however important differences stemming from the fact that this research line is interested in inducing latent representations to be used in tasks such as image generation, whereas the goal of emergent communication is to induce discrete symbols for inter-agent coordination. As a result, for example, discrete representations derived with variational autoencoders are typically of much higher dimensionality, making a direct comparison in terms of interpretability difficult. Still, it is remarkable that the goal of learning discrete representations in an unsupervised way independently arose in different fields, and future work should explore connections between these ideas.

**Self-supervised representation learning** Self-supervised learning of general-purpose visual features has received much attention in recent years. The main idea is to train a network on a pretext task that does not require manual annotation. After convergence, the net is used to extract high-quality features from images, to be applied in various "downstream" tasks of interest. This is often done by training a simple classifier on top of the frozen trained architecture [23–26].

Early models used image-patch prediction as the proxy task [8, 27]. Recent work has instead focused on an instance-level contrastive discrimination objective [5–7, 28]. Two symmetric networks encode different views of the same input images obtained through a stochastic data augmentation pipeline. Optimization is done with variants of the InfoNCE loss [27, 29], that maximizes similarity among representations of the same image while minimizing similarity of different ones.

Interestingly, the contrastive pretext task is very close to the one of identifying a target image among distractors, as in the standard emergent communication referent discrimination game. The

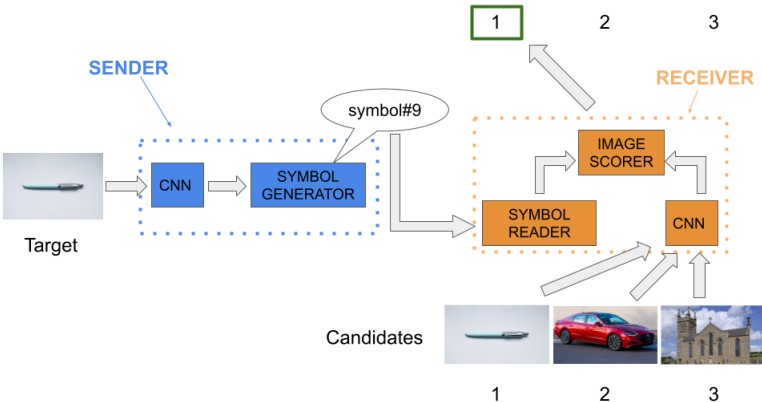

Figure 1: Game setup and agent architectures. Image sources: `https://unsplash.com`

influential SimCLR model proposed by Chen et al. [5] is particularly similar to our setup. It uses two twin networks with a shared convolutional module optimizing the (dis)similarity of sets of target/distractor images. The main conceptual differences are that there is no discrete bottleneck imposed on "communication" between the networks, and there is no asymmetry, so that both networks act simultaneously as Sender and Receiver (both networks produce a continuous "message" that must be as discriminative for the other network as possible).

We rely here on the connection with self-supervised learning in two ways. First, we import the idea of data augmentation from this literature into the communication game, showing how it helps in evolving a more semantically interpretable protocol. Second, we evaluate the discrimination game as a self-supervised feature extraction method. We find that the visual features induced by the CNNs embedded in our agents are virtually as good as those induced by SimCLR, while the emergent protocol is better for communication than the one obtained by adapting SimCLR to the discrete communication setup.

## 3 Setup

### 3.1 The discrimination game

A Sender network receives as input a *target* picture, and it produces as output one of $|V|$ *symbols*. A Receiver network receives in input this symbol, as well as a list of $n$ pictures, one of them (randomly placed in the $i$th position of the list) being the same target presented to the Sender. Receiver produces a probability distribution of cardinality $n$, interpreted as its guess over the position of the target. The guess is correct iff Receiver concentrates the largest probability mass on the $i$th position, corresponding to the target slot.

**Agent architecture** Agent architecture and game flow are schematically shown in Fig. 1. Sender reads the target image through a convolutional module, followed by a one-layer network mapping the output of the CNN onto $|V|$ dimensions and applying batch normalization [30], to obtain vector $\mathbf{v}$. Following common practice when optimizing through discrete bottlenecks, we then compute the Gumbel-Softmax continuous relaxation [31, 32], which was shown to also be effective in the emergent communication setup [11]. At train time, Sender produces an approximation to a one-hot symbol vector with each component given by $m_i = \frac{\exp\left[(s_i+v_i)/\tau\right]}{\sum_j \exp\left[(s_j+v_j)/\tau\right]}$, where $s_i$ is a random sample from Gumbel(0,1) and $v_i$ a dimension of $\mathbf{v}$. The approximation is controlled by temperature parameter $\tau$: as $\tau$ approaches 0, the approximation approaches a one-hot vector, and as $\tau$ approaches $+\infty$, the relaxation becomes closer to uniform. Importantly, at test time the Sender's output is generated by directly argmax-ing $\mathbf{v}$, so that it is a discrete one-hot vector indexing one of $|V|$ possible symbols.

Receiver passes each input image through its visual module (a CNN architecture), followed by a two-layer MLP with batch normalization and ReLU after the first layer [33]. It then computes temperature-weighted cosine scores for the linearly embedded symbol compared to each image

representation. The resulting vector of cross-modal (symbol-image) similarities is transformed into a probability distribution over which image is the likely target by applying the softmax operation.

For both Sender and Receiver, we use ResNet-50 [34] as visual module. As they are different agents, that could (in future experiments) have very different architectures and interact with further agents, the most natural assumption is that each of them does visual processing with its own CNN (no weight sharing). We consider however also a setup in which the CNN module is shared (closer to earlier emergent-communication work, where the agents relied on the same pre-trained CNN).

**Optimization**   Optimization is performed end-to-end and the error signal, backpropagated through Receiver and Sender, is computed using the cross-entropy cost function by comparing the Receiver's output with a one-hot vector representing the position of the target in the image list.

**SimCLR as a comparison model**   Given the similarity between the referential communication game and contrastive self-supervised learning in SimCLR [5], we use the latter as a comparison point for our approach. Fig. 2 schematically shows the SimCLR architecture. The crucial differences between SimCLR and our communication game are the following: i) In SimCLR, the agents are parameterized by the same network, that is, the visual encoder and transformation modules in the two branches of Fig. 2 are instances of the same net. ii) The setup is fully symmetric. Like our Receiver, both agents get a set of images in input, and, like our Sender, both agents can be seen as producing "messages" representing each input image. iii) Instead of (a probability distribution over) symbols, the exchanged information takes the form of continuous vectors ($s$ in the figure). iv) The loss is based on directly comparing embeddings of these continuous vectors ($z$ in the figure), maximizing the similarity between pairs representing the same images (positive examples in contrastive-loss terminology) and minimizing that of pairs representing different images (negative examples). This differs from our loss, that maximizes the similarity of the Receiver embedding of the Sender-produced discrete symbol with its own representation of the target image, while minimizing the similarity of the symbol embedding with its representation of the distractors.

It is important to stress the different roles that SimCLR as a comparison model will play in the experiments below. When playing the communication game (Section 4.1), our discretized Sim-CLR method must be simply seen as an interesting baseline, as the system was not designed for discrete communication in the first place (it is indeed interesting that it performs as well as it does). When performing protocol analysis (Section 4.2), the discrete clusters explicitly built on SimCLR representations act as a challenging comparison point for the categories implicitly induced by our system through game playing. Finally, when evaluating the visual features learnt by the two systems (Section 4.3), the roles are inverted, with SimCLR being a standard method developed for these purposes, whereas in our setup the emergence of good visual representations is a by-product of communication-based model training.

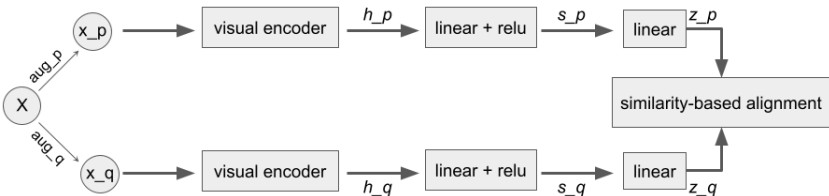

Figure 2: The SimCLR architecture. *aug_p* and *aug_q* are outputs of the stochastic augmentation pipeline used to generate two views of the same image.

**Data augmentation**   In the original discrimination game proposed in [9], the agents are shown exactly the same target image.[2] In self-supervised learning, on the other hand, it is common practice to "augment" images in different ways, e.g., by applying different croppings or color perturbations

---

[2]Lazaridou and colleagues [9] also considered a variant of the game in which the agents see different pictures of the same category (e.g., the shared target is *dog*, but the agents get different dog pictures). This version of the game is however severely limited by the requirement of manual category annotation. Lazaridou et al. [13] also provide different images to Sender and Receiver, by feeding them different viewpoints of the same synthetically generated objects: again, a strategy that will not scale up to natural images.

[5, 35, 36]. In standard contrastive learning frameworks, where all the weights are shared and there is no communication bottleneck, it is necessary to create these different views, or else the system would trivially succeed at the pretext contrastive task without any actual feature learning. We conjecture that data augmentation, while not strictly needed, might also be beneficial in the communication game setup: presenting different views of the target to Sender and Receiver should make it harder for them to adopt degenerate strategies based on low-level image information [19]. We follow the same data augmentation pipeline as [5], stochastically applying crop-and-resize, color perturbation, and random Gaussian blurring to every image. Note that, for the experiments reported in the main paper, we do not apply data augmentation at test time. Results with augmentation also applied at test time are reported in Appendix A.4.

**Implementation details**    All hidden and output layers are set to dimensionality 2048.[3] Note that this implies $|V| = 2048$, more than double the categories in the dataset we use to train the model (see Section 3.2 below), to avoid implicit supervision on optimal symbol count.[4] We fix Gumbel-Softmax temperature at 5.0, and Receiver cosine temperature at 0.1. The latter value is also used for the equivalent $\tau$ parameter in the NTXent-loss of our SimCLR implementation.

We train with mixed precision [37] for 100 epochs, with a batch of size $16 \times 128 = 2048$, divided across 16 GPUs. Rather than sampling distractors from the entire dataset, we take them from the current device's batch, thus playing the communication game with 127 distractor images in all reported experiments.[5] We do not share distractors (negative samples) across devices. As in SimCLR, we use the LARS optimizer [38] with linear scaling [39], resulting in an initial learning rate of 2.4. We apply a cosine decay schedule without warmup nor restart [40]. Compute requirements are reported in Appendix A.1. All models are implemented with the EGG toolkit [41].

## 3.2   Data

**Referential game**    Training targets and distractors are sampled from the ILSVRC-2012 training set [42], containing 1.3M natural images from 1K distinct categories. We use two image sources for testing. First, we use the ILSVRC-2012 validation set, containing around 50K images from the same categories as the training data. Second, in order to probe the generality of the emergent protocol, we introduce a new "out-of-distribution" dataset (henceforth, the OOD set). To build the latter, we relied on the whole ImageNet database [17], exploiting its WordNet-derived hierarchy [43]. In particular, we randomly picked (and manually sanity-checked) 80 categories that were neither in ILSVRC-2012 nor hypernyms or hyponyms of ILSVR-2012 categories (e.g., since *hamster* is in ILSVRC-2012, we avoided both *rodent* and *golden hamster*). We also attempted to sample categories of comparable degree of generality to those in ILSVRC-2012. For each of the categories chosen according to these criteria, we randomly sampled 128 images from ImageNet. Examples of included categories are *eucalyptus*, *yellowtail*, and *drawer*. The OOD set categories are not tremendously different from the ones in ILSVRC-2012, belonging to similar high-level domains, such as plants, fish and furniture. This is on purpose. As our agents are limited to single-symbol communication, we do not expect them to be able to denote completely unrelated test categories by compositional means. Rather, the function of the OOD set is to check that the symbols of an emergent protocol do not overfit the very specific classes of the training set, but are general enough to be usable for somewhat related categories (e.g., that symbols developed for other training-set fish categories can also denote yellowtail).[6]

**Linear evaluation of visual features on downstream tasks**    Following standard practice in self-supervised learning [e.g., 5, 23, 44], we evaluate the visual features induced by the CNN components of our models by training a linear object classifier on top of them. We use four common data sets:

---

[3]This is the same size used in the original SimCLR paper, except for the nonlinear projection head. For the latter, a number of sizes were tested, and the authors report that they do not impact final performance. We use 2048 for direct comparability with our setting.

[4]Results with different vocabulary sizes are reported in Appendix A.6

[5]Results with different numbers of distractors are reported in Appendix A.5.

[6]Paths to the ImageNet images in the OOD set and the corresponding categories are available at `https://github.com/facebookresearch/EGG/blob/master/egg/zoo/emcom_as_ssl/OOD_set.txt`.

ILSVRC-2012, Places205 [45], iNaturalist2018 and VOC07.[7] Evaluation is carried out with the VISSL toolkit [46],[8] adopting the hyperparameters in its configuration files without changes.

## 4 Experiments

### 4.1 Referential communication accuracy

We start by analyzing how well our models learn to refer to object-depicting images through a learned protocol.[9] As an interesting baseline, we let the trained SimCLR model play the referential game by argmax-ing its $s$ layer into a discrete "symbol" ($\text{SimCLR}_{disc}$). By looking at the SimCLR diagram in Fig. 2, it should be clear that $s$ constitutes the equivalent of the communication layer, with $z$ functioning as symbol embedding layer. Discretizing $h$, in any case, led to lower game-playing performance.[10]

Accuracy is given by the proportion of times in which the Receiver "picks the right image", that is, it assigns the largest symbol-embedding/image-representation similarity to the target compared to 127 distractors (chance $\approx 0.8\%$).

Results are in Table 1. The ILSVRC-val column shows that all models can play the game well above chance when tested on new images of the same categories encountered during training. The next column (OOD set) shows that the models also play the game well above chance with input images from new categories, although mostly with a drop in performance. All variants of our model are considerably more robust than the SimCLR baseline (which, however, does remarkably well at this discrete communication game it was not designed for).

|  | ILSVRC-val | OOD set | Gaussian Blobs |
|---|---|---|---|
| $\text{SimCLR}_{disc}$ | 56.9% | 47.4% | 0.8% |
| Communication Game | | | |
| -augmentations -shared | 91.2% | 90.8% | 43.4% |
| -augmentations +shared | 92.8% | 92.7% | 84.7% |
| +augmentations -shared | 81.5% | 72.0% | 0.8% |
| +augmentations +shared | 82.2% | 73.7% | 0.8% |

Table 1: Game-playing accuracy. $\pm$ *augmentations* marks whether the game was trained with data augmentations or not. $\pm$ *shared* indicates whether there was CNN weight sharing or not between Sender and Receiver.

Looking at model variants, sharing CNN weights or not makes little difference (an encouraging first step towards communication between widely differing agents, that will obviously not be able to share weights). On the other hand, data augmentations apparently harm performance. However, it turns out that the better performance of the non-augmented models is due to an opaque communication strategy in which the agents are referring to low-level aspects of images (perhaps, specific pixel intensity levels?), and not to the high-level semantic information they contain (ideally, object categories).

To show this, we replicated the sanity check from [19]. We freeze the trained models and let them play the communication game with blobs of Gaussian noise as targets and distractors. We use 384 batches of 128 224x224-sized random images whose pixels are drawn from the standard Gaussian distribution $\mathcal{N}(0, 1)$, for a total of $49152$ items (a size comparable to that of ILSVRC-val). Results are in the last column of Table 1. The *+augmentations* models and SimCLR fully pass the sanity check, with performance exactly at the $0.8\%$ chance level. The *-augmentation* models, on the other

---

[7] http://places.csail.mit.edu/index.html, https://www.kaggle.com/c/inaturalist-2018, http://host.robots.ox.ac.uk/pascal/VOC/voc2007/

[8] https://vissl.ai/

[9] Appendix A.2 reports this and all following experiments repeated with 5 distinct initializations of our most representative model (*+augmentations -shared*). It shows that variance across runs is negligible. We did not repeat the check for the remaining models due to time and resource constraints (see Appendix A.1).

[10] We also compared to an architecture from earlier work in emergent communication with realistic visual inputs, that is, the one of [9, 19]. However, as shown in Appendix A.3, its performance is considerably lower than that of $\text{SimCLR}_{disc}$.

hand, are able to use the symbols they learned from sane input to communicate about the Gaussian blobs, showing that they developed an opaque protocol. The *-augmentations +shared* model, in particular, is hardly affected by the switch to noise data.

## 4.2 Protocol analysis: emergent communication as unsupervised image annotation

The Gaussian blob test suggests that the *+augmentations* models do not fall into the trap of a degenerate low-level protocol. However, it is not sufficient to conclude that they learned to associate symbols with human-meaningful referents. To test whether this is the case, we exploit the fact that, as we are working with ImageNet data, we have labels denoting the objects depicted in the images. We use this information in two ways. We compute the normalized mutual information (nMI) between the ground-truth labels of target images and the symbols produced for the same images by the trained Sender. The nMI of two variables is obtained by dividing their MI by their average entropy, and it ranges between 0 and 1. We also compute a normalized similarity measure based on the shortest path between two categories in the WordNet *is-a* taxonomy. Our WNsim score is the average shortest-path similarity of the ground-truth categories of all target pairs that share the same Sender symbol, and it also ranges between 0 and 1. WNSim is more nuanced than nMI, as it will penalize less a Sender using the same symbol for similar categories (e.g., cats and dogs) than one using the same symbol for dissimilar ones (cats and skyscrapers). WNsim is computed with NLTK [47].[11]

We again take SimCLR$_{disc}$ as a comparison point, where a "symbol" is simply the dimension with the largest value on a certain layer. To give this approach its best chance, we evaluated the $h$, $s$ and $z$ layers (see Fig. 2), and report statistics for $h$ (CNN output), as it produced the best overall scores across data sets. We also run $k$-means clustering on the $h$ layer (SimCLR$_{kmeans}$). Cluster centroids were estimated on a random 10% of the training set, and then used to group the test images into clusters (treated as equivalents to the symbols produced by our systems). We tried clustering with $k = 1000$ (ground-truth class cardinality) and $k = 2048$ (same as vocabulary size of our models). We report the significantly better results we obtained with the second choice.

| | ILSVRC-val | | | OOD set | | |
|---|---|---|---|---|---|---|
| | $|P|$ | nMI | WNsim | $|P|$ | nMI | WNsim |
| SimCLR$_{disc}$ | 1489 | 0.49 | 0.11 | 1069 | 0.46 | 0.19 |
| SimCLR$_{kmeans}$ | 2035 | 0.59 | 0.18 | 1519 | 0.54 | 0.30 |
| Communication Game | | | | | | |
| -augmentations -shared | 2044 | 0.50 | 0.08 | 1921 | 0.45 | 0.11 |
| -augmentations +shared | 2048 | *NS* | *NS* | 2025 | *NS* | *NS* |
| +augmentations -shared | 2042 | 0.58 | 0.18 | 1752 | 0.53 | 0.32 |
| +augmentations +shared | 2046 | 0.56 | 0.15 | 1765 | 0.51 | 0.25 |

Table 2: Protocol analysis. |P| is the observed protocol size, that is, the number of distinct symbols actually used at test time. We mark as NS the cases where the obtained scores were not significantly different from chance according to a permutation test with $\alpha = 0.01$.

Looking at Table 2, we first observe that, consistent with the Gaussian blob sanity check, there is no significant sign of symbol-category association for the *-augmentations +shared* protocol. All other models show some degree of symbol interpretability (with significantly above-chance nMI and WNsim scores). Even when there is no data augmentation during training, using different visual modules (*-augmentations -shared*) leads to some protocol interpretability, coherently with the fact that this configuration was less able than its *+shared* counterpart to communicate about noise. We were moreover surprised to find that simply argmaxing the SimCLR visual feature layer produces meaningful "symbols", which suggests that information might be more sparsely encoded by this model than one could naively assume.

Importantly, the game protocols derived with data augmentation have particularly high nMI and WNsim scores. Impressively, the scores achieved by our models, and the *+augmentations -shared* setup in particular, are very close to those obtained by clustering SimCLR visual features. Recall that, unlike our models, whose protocol independently emerges during discriminative game training, SimCLR$_{kmeans}$ runs a clustering algorithm on top of the representations produced by the SimCLR

---

[11] Appendix A.7 further reports an analysis of symbol frequency distributions for representative models.

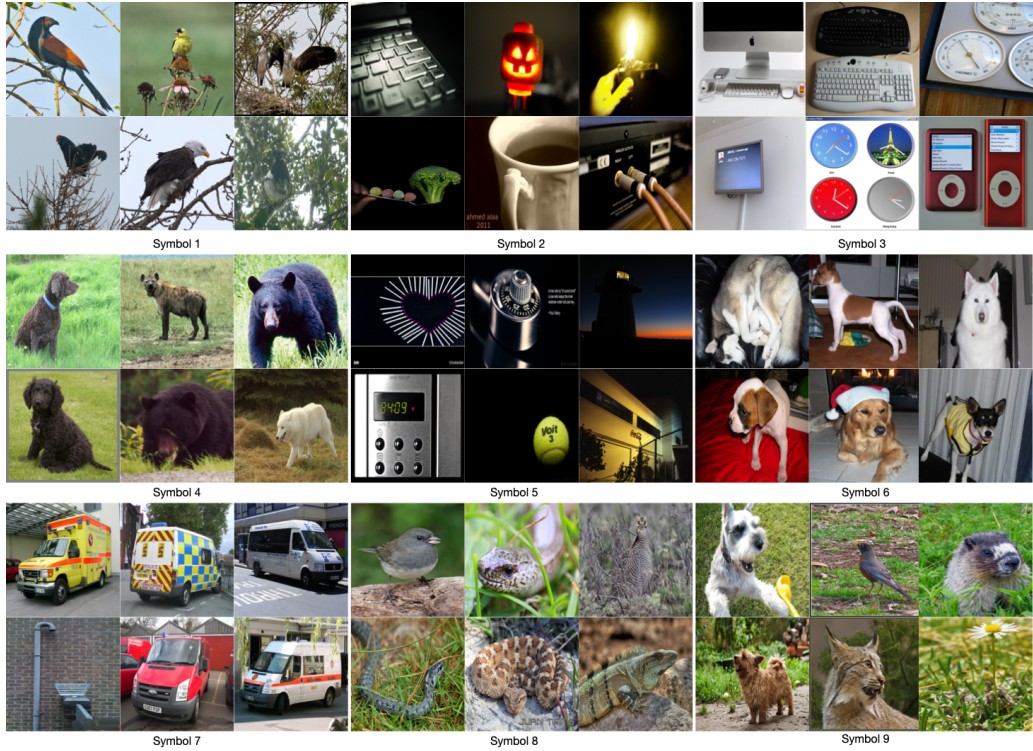

Figure 3: Randomly selected ILSVRC-val images triggering the +*augmentations -shared* Sender to produce its 9 most frequent symbols.

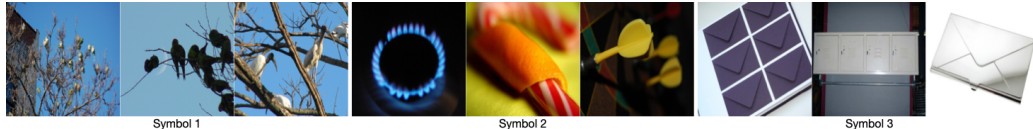

Figure 4: Randomly selected OOD set images triggering the +*augmentations -shared* Sender to produce the 3 most frequent ILSVRC-val symbols (cf. Fig. 3).

visual encoder with the express goal to discretize them into coherent sets, thus constituting a hard competitor to reach.

Beyond the quantitative results, a sense of how good our symbols are as unsupervised image labels can be gauged by qualitatively inspecting images sharing the same assigned symbol. Fig. 3 shows a random set of such images for the 9 symbols most frequently produced by the +*augmentations -shared* Sender in ILSVRC-val, without hand-picking.[12] Some symbols denote intuitive categories, although, interestingly, ones that do not correspond to specific English words (birds on branches, dogs indoors... ). Other sets are harder to characterize, but they still share a clear high-level "family resemblance" (Symbol 2: objects that glow in the dark; Symbol 3: human artifacts with simple flat shapes, etc). Frequency imbalance in input super-categories, together with the fact that the agents are allowed to use a large number of symbols, leads to partially overlapping clusters (Symbol 9 might denote living things in the grass, whereas Symbol 4 seems to specifically refer to mammals in the grass).

The fact that symbols do not exactly denote ILSVRC categories plays to the agents' advantage when they must communicate about OOD images. While this set, by construction, does not contain ILSVRC categories, as Fig. 4 shows, it still contains birds on tree, glowing objects and artifacts with

---

[12]We excluded 25% of symbol-5 images before sampling, as they depicted people. Consistent with this symbol's "theme", the latter mostly show people with dark backgrounds.

|  | ILSVRC-val | Places205 | iNaturalist2018 | VOC07 |
|---|---|---|---|---|
| Supervised | 76.5% | 53.2% | 46.7% | 87.5% |
| SimCLR | 60.6% | 49.0% | 31.8% | 78.7% |
| Communication Game |  |  |  |  |
| +augmentations -shared | 59.0% | 47.9% | 30.8% | 77.0% |
| +augmentations +shared | 60.2% | 49.1% | 31.3% | 78.8% |

Table 3: Linear evaluation on object classification. Reported scores are mAP for VOC07, top-1 accuracy elsewhere. Supervised results are from [24].

flat shapes, so that the agents can successfully refer to them by using the "spurious" but interpretable symbols denoting these concepts that they induced from ILSVRC.

### 4.3 Downstream object classification: emergent communication as self-supervised visual feature learning

Finally, we evaluate the features produced by the Sender CNN trained on the communication game as out-of-the-box visual representations. We follow the standard protocol for training a linear classifier on the output of the frozen CNN trunk on various object classification data sets (see Section 3.2 above). We focus on Sender because the features produced by the two agent networks are always highly correlated.[13] We further exclude the *-augmentations* models that, having learned a degenerate strategy, reach extremely poor classification performance on ILSVRC-val (below 5% accuracy).

As a reasonable upper bound, Table 3 reports the fully-supervised object classification results from [24]. As a more direct point of comparison, we also report the performance of our SimCLR implementation.[14] The table shows that the features developed as a by-product of the communication game are of comparable quality to those of SimCLR, a method developed specifically for visual feature learning. This is an extremely promising first step towards employing emergent communication as a form of self supervision. Many ideas from the self-supervised literature (e.g., new data augmentation pipelines, the use of memory banks for distractor sampling or variants of the similarity-based pretext task) could straightforwardly be integrated into our setup, hopefully leading to the emergence of even better visual features and, perhaps, an even more transparent protocol.

## 5 Conclusion

Deep agent coordination through communication has recently attracted considerable interest. Referential games are a natural environment to test the agents' emergent communication strategies. Past approaches, however, relied on relatively small image pools processed with pretrained visual networks, or on artificial input. We showed here that deep agents can learn to refer to a high number of categories depicted in large-scale image datasets, while communicating through a discrete channel and developing their visual processing modules from scratch. Performance on referential games with two distinct test sets (one with categories not presented at training), along with protocol analysis, shows that the agents' protocol is effective and partially interpretable.

A key ingredient to success was input data augmentation. We borrowed this idea from recent approaches to self-supervised visual learning. Conversely, we showed how the agents' visual networks emerging from discriminative game playing produce high-quality visual features. Further integration of emergent communication and self-supervised learning methods should be explored in the future.

Our work constitutes just a small step in the right direction, and has important limitations to be addressed in future work. Our agents communicate through a single symbol, but the true expressive

---

[13]Across setups and data sets, the Sender/Receiver correlation between all pairwise visual representation similarities was never below 0.96.

[14]It is difficult to compare our SimCLR ILSVRC-val performance precisely to that reported in the original paper since, coherently with the communication game setup, we use a per-GPU batch size of 128 without sharing negatives across GPUs. By looking at the leftmost bars of Fig. 9 in [5], we note that the performance we report is within the range of their results for the same number of training epochs (100).

power of human language comes from the infinite combinatorial possibilities offered by composing sequences of discrete units [48]. Allowing longer messages and probing whether this results in the development of a compositional code is our next priority. Additionally, while in our experiments distractors are selected at random, this is obviously not the case in real-life referential settings (dogs will tend to occur near other dogs or humans, rather than between a whale and a space shuttle). Dealing with realistic category co-occurrence is thus another important future direction. Finally, although we argued that a discrete protocol should be more interpretable than a continuous one, and we provided preliminary quantitative and qualitative evidence that the agents' protocol is indeed reasonably transparent, whether the achieved degree of interpretability is good enough for human-in-the-loop scenarios remains to be experimentally investigated.

Much recent work in the field has moved towards a theoretically-oriented understanding of deep agent communication in symbolic and artificial setups. We went back instead to the original motivation behind the study of emergent language, as a path towards the development of autonomous AIs that can interact with each other in a realistic environment. While we are still far from real-life-deployable interpretable machine-machine interaction, we hope that our work will stimulate more studies pursuing this ambitious goal.

## Acknowledgments

We would like to thank the NeurIPS area chair and reviewers, as well as Gemma Boleda, Rahma Chaabouni, Emmanuel Chemla, Simone Conia and Lucas Weber for feedback; Priya Goyal for technical support with VISSL; Mathilde Caron and the participants in the EViL meeting, the FAIR EMEA-NLP meetup and the Trento/Amsterdam/Barcelona (TAB) meeting for fruitful discussions. We also want to thank Jade Copet for sharing an early version of the code used in this work.

## Funding transparency statement

The authors did not receive any form of third-party funding or support. They do not have any financial relationship with entities that could be considered broadly relevant to the work, except those declared in their affiliations.

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
