# A    Appendix to Interpretable agent communication from scratch (with a generic visual processor emerging on the side)

## A.1    Compute details

All experiments were run using Tesla V100 GPUs on a SLURM-based cluster, except where indicated. Training a communication game takes approximately 16 hours on 16 GPUs. Testing on the referential game takes less than 5 minutes on a single NVIDIA Quadro GP100. The permutation procedure used to establish statistical significance for purposes of protocol analysis takes up to about 24 hours, and does not require GPUs. The downstream object classification experiments take up to about 16 hours on 8 GPUs.

## A.2    Impact of random seeds on +*augmentations* -*shared* model performance

To gauge the robustness of our results to model initialization variance, we repeated all experiments after training our most representative model (+*augmentations* -*shared*) with 5 different random seeds (including the randomly picked seed consistently used for the results reported in the main text). The outcomes, summarized in Tables 1, 2 and 3, show that the effect of this source of variation on model performance is negligible.

| Dataset | avg | sd | min | max |
|---|---|---|---|---|
| ILSVRC-val | 81.4% | 0.2% | 81.1% | 81.6% |
| OOD set | 71.7% | 0.6% | 71.0% | 72.4% |
| Gaussian Blobs | 0.8% | 0.1% | 0.8% | 1.0% |

Table 1: Game playing accuracy of +*augmentations* -*shared* model across 5 seeds.

| Dataset | avg | sd | min | max |
|---|---|---|---|---|
| ILSVRC-val | | | | |
| $\|P\|$ | 2040.4 | 2.1 | 2037 | 2042 |
| nMI | 0.58 | 0.00 | 0.58 | 0.58 |
| WNsim | 0.18 | 0.00 | 0.17 | 0.18 |
| OOD set | | | | |
| $\|P\|$ | 1749.8 | 15.0 | 1723 | 1767 |
| nMI | 0.53 | 0.00 | 0.52 | 0.53 |
| WNsim | 0.29 | 0.02 | 0.27 | 0.32 |

Table 2: Protocol analysis statistics of +*augmentations* -*shared* model across 5 seeds.

| Dataset | avg | sd | min | max |
|---|---|---|---|---|
| ILSVRC-val | 59.1 | 0.1 | 59.0 | 59.2 |
| Places205 | 48.2 | 0.2 | 47.9 | 48.3 |
| iNaturalist2018 | 31.1 | 0.2 | 30.8 | 31.3 |
| VOC07 | 77.1 | 0.1 | 77.0 | 77.2 |

Table 3: Linear evaluation accuracy on object classification for +*augmentations* -*shared* model across 5 seeds. Reported scores are mAP for VOC07, top-1 accuracy elsewhere.

## A.3    Testing the architecture of Lazaridou et al. 2017

Comparing our model to previous emergent communication architectures is somewhat problematic. Being a relatively young field, there is a limited number of earlier models it makes sense to compare against. Additionally, when employing visual input, most previous work relied on ad-hoc game configurations and architectures that were trained with a small number of classes. Most importantly, our aim here is not to compare with previous models on a specific metric or data set. Instead, we want show that, with an appropriate training setup, we can induce the emergence of large-scale communication about realistic images using generic architectures.

Putting these caveats aside, we report here results for the most directly comparable model from previous work. This is the model of Lazaridou et al. [1] and Bouchacourt and Baroni [2], in the variant that they refer to as *Informed Sender*. This model can exploit extra information as its Sender architecture has access to both target *and* distractor(s) when producing a message. First, a pre-trained visual encoder embeds the target and distractors, then a convolutional filter over the image candidates is applied by treating them as separate channels. This makes the model structurally biased towards a comparative strategy when generating symbols. In the original setup, the Informed Sender used a VGG architecure pre-trained on ImageNet as visual feature extractor. The Receiver is a standard feedforward neural network, also equipped with a pre-trained VGG as visual processor. For additional details, we refer readers to [1].

We used the ILSVRC training data and experimented with and without the data augmentation pipeline described in the main text. Similarly to our setup, we employ a ResNet-50 instead of VGC but, like Lazaridou, we pre-train it on ImageNet. All other parameters are taken from the main experiment and the setup described in Section 3.1. Results when training and testing the model with 127 distractors are presented in Table 4.[1] The Informed Sender is able to generalize above chance level (0.8%) on both the in-distribution and out-of-distribution test sets but still fares far below our generic architecture or even the SimCLR$_{disc}$ baseline (see Fig. 1 in the main text). Interestingly, the ad-hoc structure of the Informed Sender does not play to its advantage. The extra information that the Sender can exploit about the distractors does not help it generating more discriminative messages. Overall, the use of data augmentations has only a marginal impact on game accuracy. Curiously, this impact is negative, possibly due to how augmentations affect the transmission of comparative multi-image information (as opposed to a single target image description).

Interestingly, the Informed Sender, even when trained without data augmentation, passes the Gaussian test, staying at chance accuracy (0.8%). Such behaviour suggests that poor performance in the communication game is not due to a degenerate language based on describing low-level visual information. Note that the Gaussian test revealed instead the emergence of a degenerate protocol for our models trained from scratch *without* augmentations and for the original Informed Sender of Lazaridou and colleagues (as shown by Bouchacourt and Baroni [2]). We leave to future work a deeper understanding of the relation between Sender architectures and the tendency to converge on the degenerate communication strategies detected by the Gaussian test.

|  | ILSVRC-val | OOD set | Guassian Blobs |
|---|---|---|---|
| -augmentations | 31.2% | 30.9% | 0.8% |
| +augmentations | 27.7% | 27.0% | 3.6% |

Table 4: Accuracy of the communication game with the Informed Sender from Lazaridou et al. [1] trained and tested with 127 distractors.

## A.4 Applying data augmentation at test time

We consider here a version of the communication game in which data augmentations are also applied at test time. On the one hand, this is not a very realistic experiment, as there is no reason why agents should see randomly augmented images when wandering around a real-life environment. On the other, it can be considered a rough approximation to the real-world challenge that two agents will rarely get identical views of the target and distractor objects.

Results are in Table 5. As expected, performance is affected across the board. Accuracy is however still much higher than random (0.8%) for SimCLR and the communication models that were exposed to augmentations at training time, with a clear advantage for the latter. Not surprisingly, accuracy drops to random level or just above it for models that were trained without augmentations.

---

[1]The design of the Informed Sender does not allow for changes in the number of distractors between training and testing regime. For this reason we cannot train it with 1 distractor as done in the original paper, and then test it with 127, for comparison with our results.

|  | ILSVRC-val | OOD set |
|---|---|---|
| SimCLR$_{disc}$ | 42.9% | 34.7% |
| Communication Game | | |
| -augmentations -shared | 1.9% | 2.0% |
| -augmentations +shared | 0.9% | 0.7% |
| +augmentations -shared | 65.6% | 54.4% |
| +augmentations +shared | 64.4% | 54.1% |

Table 5: Game-playing accuracy when data augmentation is applied at test time. $\pm$ *augmentations* marks whether the game was trained with data augmentations or not. $\pm$ *shared* indicates whether there was CNN weight sharing or not between Sender and Receiver.

## A.5 Training a communication game with a different number of distractors

An important feature of our training configuration is the use of a larger number of distractor images compared to previous emergent communication setups. This was inspired by work on self-supervised contrastive learning that relies on very large batches of datapoints to develop high-quality dense representations of the input data. In order to assess the impact of number of distractors on game-playing accuracy, we ran an ablation study with our most representative model, namely the +*augmentations -shared* setup. We conducted experiments reducing the number of distractors down to the extreme case of a single one, which is the standard setup in the earlier literature [e.g., 1, 3, 2]. Due to memory constraints we could not experiment with more than 127 distractors. At test time, we probe the models with 128 candidate images, which is equivalent to the experiment in the main text.

Results are reported in Table 6. Overall we see that training with fewer images in the candidate list drastically harms performance on both the in-distribution and out-of-distribution test sets, thus confirming that our shift towards a greater number of distractor images has positive impact on the agents' communication skills.[2]

| Distractors at train time | ILSVRC-val | OOD set | Guassian Blobs |
|---|---|---|---|
| 1 | 7.3% | 7.4% | 0.9% |
| 31 | 63.2% | 57.4% | 0.8% |
| 63 | 76.1% | 66.0% | 1.4% |
| 127 | 81.5% | 72.0% | 0.8% |

Table 6: Accuracy of the +*augmentations -shared* configuration in the communication game when training with a different number of distractors. At test time we feed the models with 128 candidate images).

## A.6 Impact of vocabulary size on the communication protocol

In Table 7 and Table 8 we report accuracy and protocol analysis for the communication game with different vocabulary sizes ($|V|$) using our most representative model (+*augmentations -shared*). We varied the size between 512 and 4096. Overall, the results show that game accuracy and protocol structure are only mildly affected by changes in vocabulary size. Increasing $|V|$ has little impact on both game playing and the emergent protocol. On the other hand, decreasing vocabulary size negatively impacts performance. Surprisingly, this is the case even when $|V|$ is close to matching the number of classes in the ILSVRC training data ($|V| = 1024$). However, the drop in performance is relatively small, suggesting that our training setup is quite robust to the choice of vocabulary size.

Interestingly, we observe that while the number of used symbols ($|P|$ in Table 11) initially approximates the number of available symbols, it then reaches a plateau at about 2.5K, suggesting that this is the "natural" amount of symbols that agents would use to describe the training set.

---

[2]It might be considered unfair to evaluate models that were trained with fewer distractors by presenting them with 128 items at test time. However, in experiments not reported here, we saw that the model trained with 127 distractors had the best overall performance even when tested with fewer distractor images, e.g., when testing models on target discrimination with a single distractor, overall accuracy was of 87.2% vs 99.4% on the OOD set for the model trained with 1 and the model trained with 127 distractors, respectively.

| Vocabulary Size | ILSVRC-val | OOD set | Gaussian |
|---|---|---|---|
| 512 | 68.8% | 62.9% | 0.8% |
| 1024 | 76.3% | 66.3% | 0.8% |
| 2048 | 81.5% | 72.0% | 0.8% |
| 3072 | 81.5% | 70.6% | 0.9% |
| 4096 | 77.7% | 66.7% | 0.8% |

Table 7: Game-playing accuracy of the *+augmentations -shared* setup with different vocabulary sizes.

| Vocabulary Size | ILSVRC-val | | | OOD set | | |
|---|---|---|---|---|---|---|
| | $|P|$ | nMI | WNSim | $|P|$ | nMI | WNSim |
| 512 | 512 | 0.48 | 0.12 | 509 | 0.42 | 0.20 |
| 1024 | 1023 | 0.53 | 0.15 | 960 | 0.50 | 0.25 |
| 2048 | 2042 | 0.58 | 0.18 | 1752 | 0.53 | 0.32 |
| 3072 | 2573 | 0.58 | 0.18 | 1711 | 0.55 | 0.30 |
| 4096 | 2461 | 0.58 | 0.17 | 1518 | 0.54 | 0.30 |

Table 8: Protocol analysis of the *+augmentations -shared* setup with different vocabulary sizes.

### A.7    Symbol distribution analysis

The histograms in Fig. 1 show the strikingly different symbol frequency distributions of the SimCLR$_{disc}$ and *+augmentations -shared* systems, when fed all ILSVRC-val images as input. For SimCLR$_{disc}$, we observe a very skewed distribution, with its mode at 1, and a few extremely frequent symbols (the most common one is used for 826 images). This indicates a strong discrepancy with respect to the underlying ILSVRC-val class distribution, which is fully balanced, with each class instantiated 50 times. The distribution emerging by playing the communication game in the *+augmentation -shared* setup (but the result also holds for other settings) is much more balanced, with mode symbol usage at 10, and the most frequent symbol being used 135 times. Comparing this to the underlying ILSVRC-val class distribution suggests that the agents generally agreed upon a more granular partition of the concept space (as symbols overwhelmingly denote sets of less than 50 images), although symbol and ground-truth label extensions are in the same order of magnitude.

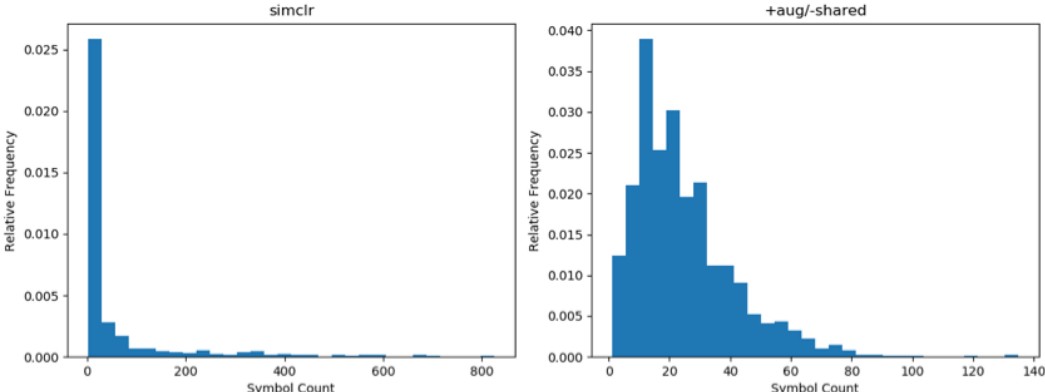

Figure 1: Histograms of relative symbol count frequencies for SimCLR$_{disc}$ (left) and the *+augmentations -shared* setup (right), given all ILSVRC-val images as input.