# OpenReview forum: "Interpretable agent communication from scratch (with a generic visual processor emerging on the side)"
_NeurIPS.cc/2021/Conference — NeurIPS 2021 Poster_

### Official Review · Reviewer_rUAU · 2021-07-14

**Rating:** 5
**Confidence:** 3

**Summary:**

This paper studies the visual features emerging from a discrete referential game between networks. A sender network processes the input image into a discrete symbol; the receiver is tasked to identify the input image from discrators images using the encoded discrete symbol. The authors study the emerging features based on their referential accuracy, interpretability, and quality as unsupervised pre-trained features.


**Limitations And Societal Impact:**

Discussed in the paper.

**Main Review:**

=== Strengths ===

+ The paper is well written and the presentation is easy to follow

+ I appreciate the analysis on interpretability in section 4. The WNsim score computed using WordNet taxonomy shows that the emergent features are semantically meaningful. The analysis on the sparsity of SimCLR features, demonstrated by its interpretability scores, is particularly useful.

+ The downstream object classification results in Table 3 are competitive against the strong baseline SimCLR. This shows great potential for the proposed referential as an alternative self-supervised pre-training method.

=== Weaknesses ===

- One of the main contributions the authors claim is that the visual features emergent from the referential game is of high quality. The authors argue this by showing competitive numbers against SimCLR in downstream classification, both with frozen features. I would like to also see the comparisons on fine-tuned features, which measure the usefulness of the emergent features more directly.

- I am confused on why there is no data augmentation during testing in the referential game (L181-182). This means the exact same image is present in the candidate images. Wouldn’t this mean that any deterministic network acting as both the sender and receiver, even with completely randomized weights, can achieve 100% accuracy, assuming there are no ties?

- The authors claim that this paper provides evidence that communication protocols emerge in a realistic setting (L8-10). In my opinion (my opinion only), this is slightly sensationalist.
Firstly, I disagree that the emergence feature can be considered as “communication protocols”, as they merely describe the implicit class of the input image. Admittedly, they are emergent and discrete. However, prior work has already shown that it is possible to learn compact discrete latent variables of images, without any supervision [1,2]. Secondly, I disagree that the described referential game is a realistic setting, as the receiver agent is only tasked to identify **the exact same image** that the sender encodes.


[1] Neural Discrete Representation Learning, van den Oord et al., 2017

[2] Generating Diverse High-Fidelity Images with VQ-VAE-2, Razavi et al., 2019


**Time Spent Reviewing:**

7

---

> ### Author Response · Authors · 2021-08-09
> **General response/experiments with data augumentation at test time**
>
> Thanks for your feedback. We will incorporate it in the paper revision.
>
> *Full fine-tuning of SimCLR and emergent communication features*
>
> This would be an interesting experiment, but we cannot run it in time for rebuttal, as it is very time consuming. We will add it to the revision, if the paper is accepted. Our survey of the SSL literature, however, suggests that the frozen feature evaluation we conduct is a good predictor of system quality in further downstream tasks.
>
> *Why no data augmentation during testing? Doesn’t this make the task trivial for a deterministic random network acting both as sender and receiver?*
>
> We preferred to report test results without augmentation as real-world input would not be randomly altered by the augmentation processes we apply during training.
>
> Due to the presence of the discrete bottleneck, the random receiver network you describe has no way to map the information it gets (a discrete symbol) to a representation matching the right image (we sanity-checked this, and indeed a random system behaves randomly).
>
> Results with data augmentation during testing are as follows: SimCLR and +augmentation models show a drop in performance, but still have high accuracies, with the +augmentation communication systems still superior to SimCLR. The -augmentation models, not surprisingly, drop to random performance. We will report the relevant table in an appendix.
>
> *Relation to earlier work mapping to discrete latent variables*
>
> Thanks for the references, that we will discuss as related work. The discrete variables produced by VG-VAE and by our communication-based approach are very different. For example, when processing images, van den Oord’s model generates 1024 spatially-encoded latent variables, vs. a single symbol in our case. This makes a direct comparison problematic, and we’d prefer to leave it to future work.
>
> *Emergent symbols just describe implicit class image, they are not “communication protocols”*
>
> The emergent symbols carry out the core referential function that a communication protocol should have, namely allowing the Receiver to identify the item that the Sender is referring to. We thus consider them as a limited but veritable form of communication protocol. We will clarify in the paper the sense in which we use the term.
>
> *Referential setting is not realistic because sender and receiver see identical images*
>
> We agree that more realistic experiments should be conducted in setups where Sender and Receiver see objects from different perspectives. Unfortunately, we are not aware of appropriate, non-synthetic data-sets for this experiment. Following your advice, we will also report results with test-time augmentations (see above). Although the distortions brought about by such augmentations are not “realistic”, they address the issue of how communication accuracy changes when the Sender and the Receiver look at different views of the same object.

---

> > ### Comment · Reviewer_rUAU · 2021-08-10
> > **Thanks for the response. Here are my follow-up questions**
> >
> > Thank you for the response. Here are my follow-up questions:
> >
> >
> > * _Referential setting, augmentation at test time_
> >
> > Thanks for the response and running the additional experiments. I believe augmenting the images during test time is more realistic as the task is in general harder. I do still have concerns on the previous setup, which I express below.
> >
> > * _Data augmentation_
> >
> > Thank you for the response. I might still be missing something here: given a deterministic network and two identical input images, wouldn’t the output of the network (acting both as the sender and receiver) also be the same? Therefore, assuming there are no ties after discretization (which is a function of choice, for example, one can simply use the argmax), such a referential game would be trivial. If a random deterministic network behaves randomly, does that mean there are a lot of ties after discretization? If this is the case, could this be due to the fact that the intermediate feature is not getting normalized? Since |V|=2048 whereas B=128, I would imagine collisions/ties happen rarely assuming regular distribution among the distracting images, especially if one uses batch normalization. I would like to get some further clarification on that.
> >
> >
> > * _Relation to earlier work mapping to discrete latent variables_
> >
> > I agree with the authors that the presented approach is different, as it is designed for a different task whereas VQ-VAE is designed to reconstruct images. I disagree, however, that the difference is substantial enough to make it incomparable. VQ-VAE outputs spatially-encoded embeddings likely because it is tasked to reconstruct images. I would imagine one can change the encoder CNN architecture (more specifically, the striding factor), and has VQ-VAE output one latent vector with K=|V|=2048, using the same VQ technique. For fairness, the baseline SimCLR that the presented approach compares to is also designed towards a different task.

---

> > > ### Author Response · Authors · 2021-08-10
> > > **Response to follow-up questions**
> > >
> > > Thanks again for constructive feedback!
> > >
> > > - *Data augmentation*
> > >
> > > To clarify, do you have in mind a system in which the image processing CNN has random weights shared by Sender and Receiver, but the symbol-emitting MLP on top of the Sender CNN and, more importantly, the symbol-embedding MLP on top of the Receiver CNN are trained in the discrimination game?
> > >
> > > We expect indeed that such system would reach high training accuracy. However, it would not be able to generalize to test images that were not seen at training time, as it would not be able to extract any meaningful information from the random image-representing vectors (they would essentially act as distinct indices without any similarity structure).
> > >
> > > If instead you mean a system where the symbol-embedding MLP of the Receiver is not trained in any way, we do not see how such Receiver could find, for each random symbol, to which image it corresponds (and, indeed, we tested a system with random vision components, and it does not go beyond chance guessing level).
> > >
> > > - *Relation to earlier work mapping to discrete latent variables*
> > >
> > > Thanks for your suggestion: it would be very interesting to compare our method to get discrete symbols to the one, based on VQ, you suggest. We will definitely mention this in future work, and we would love to experiment with it in the near future (both in the AE setting, and adapting it to the discrimination setup, as an alternative technique to learn symbols).
> > >
> > > The SimCLR baseline is indeed "unfair" when used for the discrimination game and in terms of interpretability (where the interesting result, to us, is actually how well it works!), but it does constitute a challenging comparison point in the feature learning setup.

---

> > > > ### Comment · Reviewer_rUAU · 2021-08-10
> > > > **some clarifications**
> > > >
> > > > Thank you for the response.
> > > >
> > > > * Data augmentation
> > > >
> > > > What I have in mind is a standard deterministic CNN that maps images to embeddings of size 2048. An argmax layer than converts the embeddings to a one-hot discrete symbol of 2048. Therefore, as both sender and receiver, the network would map the same image to the same discrete symbol. Now, assuming no ties/collisions, i.e. the distractor images are mapped to different discrete symbols, which is reasonable to assume given |V|=2048 >> B, the network would trivially solve the no augmentation version of the referential game, even though the network weights are random.

---

> > > > > ### Author Response · Authors · 2021-08-10
> > > > > **Thanks for further clarifying**
> > > > >
> > > > > So, what you have in mind is a receiver that i) gets the symbol produced by the Sender; ii) gets the B input images and for each of them generates a symbol; iii) picks the image for which the symbol it generated is identical to that passed by the Sender (throwing a coin in case of ties). Yes, such a network would trivially solve the task without augmentations.
> > > > >
> > > > > We would not worry too much about such system, as it would not develop other useful properties: for example, it would obviously not produce good visual features, nor symbols that group together related objects.
> > > > >
> > > > > However, we see your point and we will also add the results with augmentations at test time to the paper (as we mentioned, performance is overall a bit lower, but the general pattern doesn't change for SimCLR and the +augmentation models).

---

> > ### Author Response · Authors · 2021-08-19
> > **further issues?**
> >
> > Dear Reviewer rUAU,
> > Thanks again for your constructive feedback and for engaging in further discussion. Do you feel that, by adding the experiments with augmentations at test time and with the explanations provided in the response (and that we are adding to the paper), we have addressed your main concerns, or are there further issues we should address?
> > Best regards,
> > The Authors

---

### Official Review · Reviewer_gFT2 · 2021-07-16

**Rating:** 8
**Confidence:** 4

**Summary:**

This paper investigates a large-scale emergent communication task, where a speaker and listener agent learn to communicate in a referential game, and draws connections between emergent communication setups and recent advances in self-supervised learning (e.g. SimCLR).

The authors show that it is possible to train emergent communication agents to such a degree that the learned communication protocol is interpretable, generalizes to unseen object categories, and most surprisingly, learns visual representations that are comparable to SimCLR.

One of the key insights enabling such a result is borrowed from the contrastive learning literature: the authors propose using standard data augmentation techniques to make the speaker and listener see different "views" of the same image, to prevent overfitting to low-level visual features present in the input. This is a common concern of emergent communication systems. They convincingly show via Gaussian blob experiments that such augmentation applied at scale prevents degenerate communication schemes, and only with such an augmentation are such high-quality visual representations learnable.

The setup is a fairly typical one used in the emergent communication literature: speaker and listener are parameterized by (possibly seperate) convolutional networks, and the speaker learns to embed the identity of a target image into a one-hot vector, which is given to a listener which must identify the referred target. The entire system is end-to-end differentiable.

Overall, this paper, with its striking finding that the visual representations learned by emergent protocols are as good as SimCLR, makes an exciting step towards more seriously considering  emergent communication-based setups as an alternative way to learn good visual (and linguistic) representations for a variety of downstream tasks.

**Limitations And Societal Impact:**

**No**: In L457 of the checklist, I would caution against authors claiming that a discrete channel for multi-agent communication is *inherently* "interpretable" and will unequivocally "have a positive impact in terms of explainable AI." In fact there are many cases in the literature (e.g. Kottur et al., 2017) where such discrete protocols are distinctly uninterpretable, and care must be taken not to naively assume otherwise. Obviously encouraging more interpretable protocols, e.g. via the data augmentation methods described here, is a good step, but does not guarantee interpretability and positive impact. I'd encourage the authors to consider this more carefully either in the main text or in the supplement.

**Main Review:**

## Strengths

- Overall, this paper is very good: it presents a clear and highly scalable emergent communication setup, at a larger scale that has been attempted before, and shows that agents are able to communicate with an interpretable protocol that generalizes to unseen categories. There are a lot of experimental details given in the paper which should facilitate reproducibility of the presented results, and the released codebase will hopefully make this easy.
- The most striking finding is that the visual features learned by agents in this game are able to perform on par with SimCLR (though this is a reimplementation, which doesn't achieve quite as high performance as the best model in the original paper).
- Theoretically, I appreciate the new connections between recent self-supervised learning techniques (SimCLR) and emergent communication-based setups. I especially like the authors idea of casting SimCLR as a two-player game. Drawing these connections allow us to bring many of the insights from self-supervised learning to bear on the emergent communication problems.
- Along these lines, one of the notable borrowed insights is the idea of using data augmentation to define different "views" of an image given to speaker and listener. This greatly increases robustness of the learned communication and visual representations, which is a very common concern in the emergent communication literature (Bouchacort and Baroni 2018, Kottur et al. 2017), and (as noted in Footnote 1) has typically been ameliorated with handcrafted, non-scalable solutions. The Gaussian blobs experiments are very astute inclusions here.
- Even the naive casting of SimCLR as a zero-shot "emergent communication" baseline is interesting (e.g. that the argmax is meaningful).

## Weaknesses

- The paper does not have much significant technical novelty; it seems to be a fairly straightforward extension of a now-standard emergent communication setup in the literature. I don't see this as a crucial weakness, however. The increased scale (and the interesting tricks that enable scalable training) are drawn from the papers biggest contribution, which is the connections to the self-supervised learning literature.
- It's unclear how truly OOD the collected OOD images are, and how difficult generalization to a few held-out ImageNet categories is in this setup. At least from Figure 4, the OOD images do not seem *that* OOD to me (we still have birds for Symbol 1, for example). A figure depicting sampled OOD images and their closest in-domain WordNet classes might be useful here. More interesting would be to compare generalization outside ImageNet (e.g. any of the adversarial ImageNet datasets, or other datasets entirely, e.g. the ones used for transfer in Table 3).
- The paper could compare and contrast emergent communication-based setups to other unsupervised learning methods. For example, especially in the shared CNN setting, an emergent communication task looks very similar to a discrete autoencoder (which can be used as an "unsupervised image annotation" tool, whose features can be reused for supervised learning, etc). What are the benefits of casting this as a referential game - presumably that it is much easier/more tractable to learn to distinguish between images, rather than reconstruct ones (which is a similar purported advantage of contrastive learning)?
- It may be better to report adjusted mutual information (Vinh et al., JMLR 2010; [https://scikit-learn.org/stable/modules/generated/sklearn.metrics.adjusted_mutual_info_score.html#sklearn.metrics.adjusted_mutual_info_score](https://scikit-learn.org/stable/modules/generated/sklearn.metrics.adjusted_mutual_info_score.html#sklearn.metrics.adjusted_mutual_info_score)) rather than normalized mutual information in Table 2, as the former is adjusted for chance.

## Questions

- One of the interesting findings of the Bouchacort and Baroni paper is that, even in the referential game with different images presented to both speaker and listener, the agents are still able to generalize to gaussian blobs. What is different, then, about the augmentation+ strategy presented in this paper that makes it unable to generalize to gaussian blobs? Is it simply increased scale?
- Have authors explored the effect of vocabulary size?
- What about the number of distractor images in each game?

**Time Spent Reviewing:**

3

---

> ### Author Response · Authors · 2021-08-10
> **Response with follow-up investigations**
>
> Thanks for your constructive criticism. We have implemented several of the analyses you suggested, and we will add them to the paper appendix.
>
> *Is the OOD set truly OOD? Why not use images outside ImageNet?*
>
> Median WordNet similarity of each OOD class with the nearest ILSVRC-train class is at 0.25. For intuition, the following are randomly selected OOD/ILSVRC-train pairs having this median similarity: *yellowtail/barracouta*, *electric toothbrush/perfume*, *tooth/fountain* (!!!), *satchel/wallet* and *belfry/beacon*. This confirms that OOD categories are generally not too far from training categories. Our agents, however, could not possibly generalize to completely novel categories, as their single-symbol protocol prevents them from developing compositional object descriptions. The goal of our OOD test is then to show that the agents have developed symbols that are not overfitting to the very specific categories present in ILSVRC-train/val (e.g., they might have learned a more useful word denoting large fish in general rather than a term specifically denoting barracoutas). We will clarify this in the revision.
>
> *Comparison to discrete autoencoders*
>
> Because of the non-contrastive objective, discrete autoencoders cannot be readily adapted to discriminative communication. Moreover, they are not the state of the art in unsupervised feature learning, which makes the comparison we perform with SimCLR in this domain more challenging. The most pertinent comparison would be in the domain of latent representation interpretability. However, the discrete variables produced by the two approaches are very different: e.g., the classic discrete VG-VAE model (van den Oord 2017) produces 1024 spatially-encoded latent variables when processing images, vs. a single symbol in our case. However, there are definitely important similarities and, as another reviewer suggests, the VG-VAE architecture could be modified to make it more directly comparable to ours. We will mention discrete autoencoders in related work, and we’d like to explore the relation more systematically in future work.
>
> *Adjusted MI instead of normalized MI*
>
> We originally used adjusted MI, but we later switched to normalized MI accompanied by a permutation-based significance test (something that adjusted MI alone does not provide).
>
> *Why does the Bouchacourt different-image game generalize to Gaussian blobs? Doesn’t it involve a form of data augmentation?*
>
> Thanks for this observation, that led us to an in-depth study of what causes the Lazaridou (= Bouchacourt) model to fall into the Gaussian blob trap.
>
> Surprisingly, we found that, when using our training data, their original architecture does not fall into the Gaussian trap, even when trained without augmentations and with a single distractor (as in the original papers). We found moreover that, if we replace the blank CNN in our architecture with a pre-trained and frozen model (as in Lazaridou/Bouchacourt), the system never falls into the Gaussian trap, even when reducing the number of distractors to 1. Still, all these model variations have lower nMI/WordNet Similarity than our from-scratch model with augmentations.
>
> Summarizing, when using a pre-trained CNN, it is sufficient to use a large and varied training data-set to avoid degenerate communication strategies. When training from scratch, we get overall better-behaved protocols, but augmentations are crucial to avoid degenerated strategies. We will report all this in the paper.
>
> *Effect of vocabulary size*
>
> We have now run experiments (in the +augmentations/-shared setup) with |V| set to 512, 1024, 2048 (the original value), 3072 and 4096. We found that increasing |V| hardly affects the model behaviour. Moreover, the effective number of used symbols does not grow as fast as |V| (e.g., with |V|=2048, all symbols are used, but with |V|=4096, only 2573 are effectively used). Lowering |V| leads to (slightly) lower communication accuracy and protocol interpretability. Overall, these results (that we will report in an appendix) suggest that the setup is robust to variations in |V| as long as the latter has a large value. Agents, instead, coordinate less well when they are forced to use a smaller vocabulary, even one that is better aligned with the number of ground-truth classes (1024 ≈ 1k).
>
> *Effect of number of distractors*
>
> As the earlier literature focused on single-distractor games, we have now implemented the most extreme case in which the number of distractors is reduced to 1. Agents develop a successful communication protocol, and avoid the “Gaussian” trap in this case as well. However, the emergent protocol has much lower nMI/WNsim than when using 127 distractors, suggesting that a large number of distractors helps discover a more high-level, general protocol.
>
> *SimCLR reimplementation is below original paper performance*
>
> Our reimplementation achieves the original paper performance when using its hyperparameters. As discussed in fn. 12, we cannot however use the latter in our setup. Our results are comparable to those reported in the SimCLR paper for the closest hyperparameters to ours.
>
> *Claim about interpretability and explainable AI*
>
> Point taken: we will add hedges to our claim.

---

> > ### Comment · Reviewer_gFT2 · 2021-08-19
> > **Thanks**
> >
> > Thanks to authors for the detailed response to my review. I appreciate the inclusion of the vocabulary size and distractor experiments (as also requested by DKJ7), and the Gaussian blob results are particularly interesting.
> >
> > I've read the other reviews and the responses to those reviews, and I will keep my score the same.

---

> > > ### Author Response · Authors · 2021-08-19
> > > **Thanks!**
> > >
> > > Thanks for acknowledging, we are glad you found the new results interesting: we do, too!

---

### Official Review · Reviewer_DKJ7 · 2021-07-17

**Rating:** 5
**Confidence:** 3

**Summary:**

The model aims to train a pair of two ConvNets in the setup of the so-called discrimination game -- one ConvNet (sender) can pass discrete information to the other ConvNet (receiver) to identify the same image presented to the Sender among a set of images. Generating such discrete (bottlenecked) information is learned in an end-to-end manner, which is claimed to provide a human-interpretable inter-ConvNet communication. Compared to SimCLR (Chen et al., ICML 2020), the proposed method shows better or matched performance in terms of discrimination game-playing accuracy and downstream image classification task.


**Limitations And Societal Impact:**

The paper does not address the limitations and societal impact of their work. I would suggest adding a discussion section to describe the potential limitations of the proposed model.

**Main Review:**

The paper needs to be more organized and focused. As the main task that the paper aims to solve is the communication between two deep neural networks, I would expect to see a more thorough analysis of the communication. E.g. (i) why |V| is chosen as 2048 and what would happen with larger or smaller values, (ii) sparsity or entropy of V, and/or (iii) what information V is learning.

Moreover, the proposed model is mainly compared with SimCLR, a self-supervised image representation learning method, but such a comparison does not look fair. SimCLR is mainly trained to make closer the latent representations of the same image of different views, thus simply argmax-ing SimCLR’s internal representations for computing game-playing accuracy (Table 1) seems unfair.

Plus, as summarized in Section 2.1, it looks like there is some prior work for the deep net emergent communication task, but surprisingly none of them is compared. Some prior work has already proposed to use the discrimination game and to use discrete communication protocol. As I am not closely following this field, it is very difficult to judge its novel contributions and their significance. For me, the contribution looks like training the existing model in an end-to-end manner. I would expect to see the author’s response on this.


**Time Spent Reviewing:**

4

---

> ### Author Response · Authors · 2021-08-10
> **Response with further experiments**
>
> Thanks for your comments. We have now run several follow-up experiments and analyses to address most of your points, as detailed below.
>
> *Requests for more in depth protocol analysis:*
>
> *(i) why |V|=2048 and ablating this hyperparameter*
>
> The motivation for |V|=2048 was to stay close to the original SimCLR hyperparameters, and to use a value greatly above the number of ground-truth classes in the training data (1k), so as not to bias the system (see fn 3: we will give more prominence to the explanation in the revision).
>
> We have now run experiments (in the +augmentations/-shared setup) with |V| set to 512, 1024, 2048 (the original value), 3072 and 4096. We found that increasing |V| hardly affects model behaviour. Moreover, the effective number of used symbols does not grow as fast as |V| (e.g., with |V|=2048, all symbols are used, but with |V|=4096, only 2573 are effectively used). Lowering |V| leads to (slightly) lower communication accuracy and protocol interpretability. Overall, these results (that we will report in an appendix) suggest that the setup is robust to variations in |V| as long as the latter is large. On the other hand, agents coordinate less well when they are forced to use a smaller vocabulary, even one that is better aligned with the number of ground-truth classes (1024 ≈ 1k).
>
> *(ii) analysis of V distribution*
>
> We have now conducted this analysis, finding a number of interesting patterns. For example, for the +augmentations/-shared setup, we find that median symbol usage across ILSVRC-val is 21, and no symbol greatly dominates the distribution (maximum usage count: 135, over a total of 50k distinct inputs). As each ground-truth class is represented by 50 ILSVRC-val inputs, median usage is compatible with the idea that symbols denote concepts only slightly more granular than gold labels. On the other hand, the discretized SimCLR baseline shows a more skewed distribution, with median usage count at 7 and the most frequent symbol being used 826 times, suggesting a larger divergence from the balanced gold class distribution. We will add the full analysis to the paper.
>
> *(iii) what information V is learning*
>
> Our analysis in 4.2 is meant to addres this, but we are happy to explore questions you think we are missing.
>
> *Comparison to discretized SimCLR is not fair*
>
> As the main new insight we introduce in the paper concerns the parallelism between emergent communication and contrastive learning, we find it interesting to compare to SimCLR throughout. However, the interpretation of this comparison changes from experiment to experiment. We will clarify that in 4.1 (communication game), discretized SimCLR is indeed just a baseline (it is actually surprising how well it fares!). In the other experiments (comparing the emergent protocols to SimCLR-based k-means clusters and downstream classification), SimCLR provides a challenging comparison point for our system.
>
> *Comparison to prior emergent communication work*
>
> The only existing model readily adaptable to our large-scale, realistic-image discrimination setup is that of Lazaridou et al. 2017/Baroni and Bouchacourt 2018. We have now run this comparison, considering a set of hyperparameters either taken from Lazaridou’s paper or similar to ours. We found that, when trained on the same amount of data we are using, the Lazaridou model succeeds at the discrimination game with high accuracy, and it passes the Gaussian-blobs sanity test. However, it is considerably worse than any of our models in terms of protocol interpretability (low nMI and WordNet similarity). We have similar results for a number of ablations (+/- augmentations, Lazaridou architecture trained from scratch, etc.), that we will report in an appendix.
>
> *Novel contributions*
>
> We will make them clearer in the introduction:
>
> (i) establishing a link between emergent communication and contrastive learning and using it to…
>
> (ii) introduce data augmentation in emergent communication and
>
> (iii) use emergent communication as a self-supervised visual feature learning method;
>
> (iv) demonstrating for the first time that deep nets can develop a successful (and partially interpretable) discrete emergent protocol in a large-scale, relatively realistic setup.
>
> *Discussion of potential limitations*
>
> We will extend our discussion.

---

> > ### Author Response · Authors · 2021-08-19
> > **Have we addressed your concerns?**
> >
> > Dear Reviewer DKJ7,
> >
> > Thanks again for your constructive feedback. We would like to ask you if our response, including further experiments, addresses your concerns, and which issues you feel are still open.
> >
> > Most importantly:
> > * we ran ablation studies for |V| and number of distractors (see reply to gFT2 on the latter);
> > * we ran a study of the emergent |V| distributions
> > * we included a direct comparison to a model from the prior emergent communication literature
> > * we highlighted our novel contributions
> >
> > Best regards,
> >
> > The Authors

---

### Decision · Program_Chairs · 2021-09-27

**Decision:**

Accept (Poster)

**Comment:**

Although ratings were quite divergent, I find myself in agreement with the reviewer arguing for acceptance. There are several interesting empirical findings in this work, and I find the connections being established between emergent communication and contrastive learning very interesting. I would have liked to see referential games with message lengths greater than one and experiments varying the way distractors were sampled. Though, I do appreciate these being called out explicitly as future work. Please incorporate the additional results and clarifications from your rebuttal/discussions into the camera ready version of the paper. A couple of small suggestions on my side: I think it could be smart to reduce the size of Figure 3 and use the extra space to walk the reader through the abstract connection you are making more thoroughly. The core of the argument and the setup for why comparing with SimCLR makes sense appear to be things that readers could misunderstand, and perhaps the authors can guard against this with a bit more discussion of them. Finally, thank you for your engagement with the reviewers during the rebuttal period.